# Current Development of Chemical Penetration Enhancers for Transdermal Insulin Delivery

**DOI:** 10.3390/biomedicines11030664

**Published:** 2023-02-22

**Authors:** Vaisnevee Sugumar, Maan Hayyan, Priya Madhavan, Won Fen Wong, Chung Yeng Looi

**Affiliations:** 1School of Medicine, Faculty of Health & Medical Sciences, Taylor’s University, 1 Jalan Taylors, Subang Jaya 47500, Malaysia; 2Chemical Engineering Program, Faculty of Engineering & Technology, Muscat University, P.O. Box 550, Muscat P.C.130, Oman; 3Medical Advancement for Better Quality of Life Impact Lab, Taylor’s University, 1, Jalan Taylors, Subang Jaya 47500, Malaysia; 4Department of Medical Microbiology, Faculty of Medicine, University of Malaya, Kuala Lumpur 50603, Malaysia; 5School of Biosciences, Faculty of Health & Medical Sciences, Taylor’s University, Subang Jaya 47500, Malaysia

**Keywords:** diabetes, insulin, transdermal, chemical enhancers, deep eutectic solvents, ionic liquid, nanoparticles, emulsions, peptides

## Abstract

The use of the transdermal delivery system has recently gained ample recognition due to the ability to deliver drug molecules across the skin membrane, serving as an alternative to conventional oral or injectable routes. Subcutaneous insulin injection is the mainstay treatment for diabetes mellitus which often leads to non-compliance among patients, especially in younger patients. Apart from its invasiveness, the long-term consequences of insulin injection cause the development of physical trauma, which includes lipohypertrophy at the site of administration, scarring, infection, and sometimes nerve damage. Hence, there is a quest for a better alternative to drug delivery that is non-invasive and easily adaptable. One of the potential solutions is the transdermal delivery method. However, the stratum corneum (the top layer of skin) is the greatest barrier in transporting large molecules like insulin. Therefore, various chemical enhancers have been proposed to promote stratum corneum permeability, or they are designed to increase the permeability of the full epidermis, such as the use of ionic liquid, peptides, chemical pre-treatment as well as packaging insulin with carriers or nanoparticles. In this review, the recent progress in the development of chemical enhancers for transdermal insulin delivery is discussed along with the possible mechanistic of action and the potential outlook on the proposed permeation approaches in comparison to other therapeutical drugs

## 1. Introduction

Diabetes mellitus (DM) is a chronic metabolic disease that is characterized by an elevated level of blood glucose resulting from defects in insulin secretion, resistance to insulin action, or both [1]. According to the International Diabetes Federation, the incidence of DM in 2019 [2] was estimated to be at 463 million people with, 4.2 million mortalities worldwide. The increase in prevalence is evident in developing countries due to the lifestyle choices [3]. However, the inequalities in acquiring healthcare lead to untreated and misdiagnosis among the low-to-middle income group, which simultaneously contributes to the rise in incidences as well as diabetic complications [4]. DM itself is a broad category that substantially is classified into two common groups, namely; type one diabetes mellitus (T1DM), identified by insufficient insulin secretion, and type two diabetes mellitus, which is characterized by the presence of insulin resistance with an inadequate compensatory increase in insulin secretion [5,6]. 

The improper management of DM could potentially lead to the development of various macro- and micro-vascular diseases [7], metabolic acidosis [8], or death [9]. Therefore, it is crucial to ensure that the glycemic level in the body is controlled. That said, the common method of treatment in patients with DM, specifically in T1DM, is through subcutaneous insulin injection [10]. They typically require an insulin dosage of 0.5 to 1.0 units per kg daily [11] which at times could be achieved only with multiple injections of insulin in a day [12,13]. The subcutaneous insulin injection is often times self-administered after proper education and self-management in cases of possible dose adjustments based on the observed glucose reading [14]. It is certain that the use of insulin injection, especially if it’s required multiple times in a day, is rather invasive and painful, which often results in low compliance among patients [15,16,17]. Moreover, frequent administration may have serious repercussions such as infection, nerve damage, overdosing as well as lipohypertrophy (LH) [18,19,20]. Alternatively, patients may opt for an insulin pump for continuous subcutaneous insulin infusion, but this method still requires major technological improvements before full-fledged clinical use. Hence, these limitations quest for effective and less invasive techniques for the delivery of insulin. 

As of now, numerous research has been carried out to establish a non-invasive approach for the delivery of insulin which includes oral, intravenous, nasal, and transdermal routes [21,22,23,24,25,26]. While each of the methods may come with its own advantages and drawbacks (Table 1), transdermal delivery possesses several benefits over other routes. As such, the transdermal route includes being less invasive compared to needles [27], potential long-term release of drugs which may reduce the frequency of dosing [28], and the ability to avoid first pass effect in the liver as well as enzymatic degradation in the digestive tract [29]. Additionally, drug administration through the transdermal route can be terminated upon removal of the patch from the site at any time. Most importantly, the transdermal administration could be easier to use, and this may increase compliance among patients in order to control their glycemic levels [30]. 

However, penetration of insulin across the skin membrane remains a major challenge due to its large molecular size and low diffusion rate hindering its bioavailability [31]. In fact, insulin has a molecular weight of 5808 Daltons (Da) and contains 51 amino acids. A study demonstrated that proteins with low molecular weight (<500 Da) could easily penetrate the skin, while the passive transport of proteins with higher molecular weight, such as insulin (>5000 Da), is largely restricted [32]. To overcome the skin barriers to transdermal insulin delivery, various permeation enhancers have been developed to facilitate insulin into the skin as well as to increase bioavailability in systemic circulation. This review explores recent advances in the transdermal insulin delivery systems using chemical enhancers such as ionic liquids (ILs), nanoparticles (NPs), deep eutectic solvents (DESs) carriers, emulsions, as well as pre-treatments. Moreover, a few examples of other successfully delivered therapeutical drugs using chemical enhancers were also briefly discussed in order to compare the possible action or mechanism that could potentially shed insight into the transdermal delivery of insulin. Potential challenges and limitations for clinical applications are also discussed. 

## 2. The Skin Structure and How This Affects Insulin Penetration

Anatomically, the skin can be divided into three distinct layers, namely the epidermis, dermis, and hypodermis. In particular, the epidermis layer consists of the stratum corneum (SC), stratum granulosum, stratum spinosum, and the basal layer. SC is 10–15 µm thick in humans and is made up of keratin-filled, metabolically inactive corneocytes surrounded by a lipid matrix organized largely in bilayers. In the outermost layer of the epidermis, SC helps to form a protective barrier against harsh elements like UV rays, pollutants, microbes, and chemicals. Hence, chemical enhancers used to promote transdermal delivery of insulin must first target the stratum corneum, as this is the primary barrier to transporting insulin into the skin and, subsequently, blood circulation [31,33,34,35]. Below the SC is the viable epidermis, which measures 50–100 µm thick and is densely filled with keratinocytes and other living cells surrounded by an aqueous extracellular matrix.

Permeation of drugs into the skin can occur through three different pathways, which is the intercellular pathways (permeation through the lipid bilayer), transcellular (permeation through the hair follicle or sebaceous glands), and intracellular (permeation across the corneocytes) (Figure 1) [36].

The use of the transcellular pathway is often time observed in small hydrophilic molecules and intercellular routes for lipophilic molecules [37]. Upon the penetration of molecules across the SC, they will then diffuse through the viable epidermis and the upper portion of the dermis, followed by the absorption to the capillary bed, which is found just below the dermal-epidermal junction (Figure 1). This junction features a basal lamina, which is known to limit the transport of macromolecules, especially those larger than 40 kDa in size [38]. There is also evidence suggesting that tight junctions in the viable epidermis may pose as a permeation barrier beside the SC [39]. Since most studies demonstrate that SC is the rate-limiting step for the delivery of drugs or therapeutics, there has been relatively little attention given to the role of viable epidermis and dermis as barriers to drug delivery [40,41,42]. 

In an attempt to address these shortcomings, various chemical enhancers have been employed as a non-invasive method to deliver insulin. In short, chemical enhancers such as ILs, NPs, peptides, carriers, and skin pre-treatments have been well-studied for use in the transdermal delivery of insulin. Mechanistically, these enhancers are linked to their ability to interact with the lipid bilayer of the skin, allowing rapid penetration of insulin across membranes (Figure 2) [43]. On the other hand, some enhancers are able to interact with thiols in the skin instead of disrupting the lipid layer. This interaction increases the bioavailability of active insulin and allows it to cross the skin barrier.

## 3. Chemical Enhancers

Human skin has a tightly organized cell layer that acts as a protective barrier against external extremities, including penetration by active pharmaceutical ingredients (APIs). However, the skin is an excellent target for transdermal delivery due to its large surface area and easy access [44]. That said, for efficient and effective transdermal delivery, APIs must be able to penetrate the stratum corneum without causing permanent damage to the skin. With this in mind, researchers are fond of using chemical penetration enhancers because they are non-invasive and can be designed to cause temporary disruption of the skin barrier [45]. As such, numerous chemical enhancers with different functions have been used over the past 50 years, including DES, ILs, peptides, NPs, and dimethyl sulfoxide (DMSO) (Figure 3). In general, studies have shown that penetration-enhancing capacity is related to the interaction of chemical enhancers in disrupting lipid bilayers by extraction or fluidization, thereby increasing drug solubility and distribution across the stratum corneum [45,46,47]. Of note, the transdermal mechanism of chemical enhancers may differ between different classes of enhancers [48,49]. As such, the various types of chemical enhancers and their potential mechanisms are discussed below (Section 3.1, Section 3.2 and Section 3.3).

### 3.1. Transdermal Insulin Delivery

#### 3.1.1. Ionic Liquids 

ILs have attracted significant interest in the pharmaceutical industry as compared to existing organic solvents due to several key advantages, which include adjustable properties such as low volatility, hydrophobicity, conductivity, good solvating interactions with organic and inorganic compounds as well as high thermal and chemical stability [50,51,52]. Recently, scientists have explored the use of ILs to address issues such as low solubility and permeability of therapeutics [53,54] and improve drug bioavailability and stability [55]. For example, choline bicarbonate and geranic acid (CAGE) based ILs (Figure 4) have been developed for transdermal delivery of insulin [56]. 

It was found that the use of topical administration of insulin-CAGE actually increased the permeability of insulin through the skin via extraction of lipids from the stratum corneum. This study highlights that CAGE did not interfere with insulin function during and after transdermal delivery [22,56]. Furthermore, another study designed to understand the variability of CAGE on skin penetration of insulin elucidated the importance of the cation–anion ratio in the development of ILs. It was observed that the ratio 1:2 and 1:4 of choline bicarbonate and geranic acid demonstrated marked insulin permeability, which was thought to be due to the presence of excess geranic acid. However, skin treated with geranic acid alone failed to deliver insulin through the stratum corneum (SC). This suggests that IL formation plays an important role in enhancing insulin permeability [22,28]. On the other hand, an IL-in oil microemulsion (MEFs) consisting of choline and fatty acids made out of three different fatty acids (C18:0-stearic acid, C18:1-oleic acid, and C18:2-linoleic acid) was chosen for its high biocompatibility and low toxicity. MEFs that utilize choline fatty acids ([Chl][FAs]) significantly facilitate insulin through the SC via an intercellular pathway, where they exert a fluidizing effect on the lipid bilayer and alter the lamellar structure of SC lipids. (Figure 2A) [57]. Moreover, the use of ILs has also been demonstrated to deliver insulin in a controlled and sustained manner [22,28,55], achieving long-term glycemic control, which further demonstrates the potential of ILs in promoting transdermal insulin delivery. In addition, cholinium-amino acid-based ionic liquids have been reported to have protein-stabilizing potential. However, some of those that are effective in suppressing insulin aggregation have shown higher levels of toxicity [58]. Although the use of ILs may contribute to insulin stability and transdermal potency, the interaction between insulin and selected ILs should be further studied to avoid toxicity as well as low insulin bioavailability upon administration. 

#### 3.1.2. Skin Pre-Treatment 

The main limitation in the transdermal delivery of large therapeutic peptides is the unique arrangement of the skin [59]. Hence, this requires an enhancer that may temporarily disrupt the SC layer of the skin in order to facilitate the delivery of peptides. A few studies have shown that pre-treatment of the skin prior to insulin administration may substantially increase the bioavailability of active insulin. For instance, Sintov et al. demonstrated the possibility of transdermal insulin delivery after topical pre-treatment with iodine (povidone-iodine ointment). Blood glucose of diabetic rats pre-treated with iodine prior to the administration of insulin showed a decrease in blood glucose within 7 h as compared to rats without pre-treatment, whereby no hypoglycemia was observed [23]. This may be due to the biodegradation of insulin by skin proteinases or due to the accumulation of insulin in its inactive form. Additionally, the authors had also reported that the inactivation of insulin after transdermal application could be attributed to the interference of various sulfhydryl compounds such as glutathione (GSH), thioredoxin, or protein disulfide isomerase (PDI) by disrupting the disulfide bond connecting the two-insulin chain. This was confirmed when compounds such as GSH and gamma glutamylcysteine (γ-GC) were significantly reduced after 2 h of iodine pre-treatment. This reflects the involvement of iodine pre-treatment that works to inactivate thiols on the skin, which would significantly increase the bioavailability of active insulin for its hypoglycemic function (Figure 2B) [23]. 

Similarly, the enhancing effect of trypsin pre-treatment on the transdermal delivery of insulin was evaluated. It was found that trypsin pre-treatment had significantly facilitated the transdermal delivery of insulin at pH 3.0, with no effect observed at pH 6.0. Results also show that a 60% reduction in blood glucose was seen after 8 h with no sign of returning to the initial blood glucose reading within 8 h. The mechanistic evaluation suggests that trypsin pre-treatment alters the SC protein structure from the alpha- to the beta-form and decreases the electrical resistance of the skin allowing the permeation of insulin into the skin (Figure 2C) [60]. The alteration of the alpha form could be associated with the rupture of the peptide and disulfide bond. Alpha helix keratin is a highly stable form of keratin. They appeared coiled in structures forming disulfide bonds or hydrogen bonds of tyrosine sidechains of keratin. They have limited exposed side chains making it difficult for interaction with water or other biomolecules. On the contrary, the beta-sheet is much softer with larger exposed sides making it easier for water molecules to intercalate between beta-sheets of keratin to interact with hydrogen bonds [61]. Thus, changes in alpha-helical structure lead to the disturbance of keratin-filled corneocyte structures, loosening and disrupting cellular protein secondary structures. Of note, the skin’s electrical resistance ensures that charged molecules cannot cross the skin barrier. As such, a decrease in electrical resistance after trypsin pre-treatment may facilitate insulin crossing into the skin. This further indicates that trypsin can potentially alter keratin-filled corneocytes by altering the lipid bilayer and reducing insulin resistance in penetrating the skin barrier [60,62]. This approach may be applicable to humans if insulin permeability and bioavailability are consistent with increased surface area. 

#### 3.1.3. Nanoparticles 

NPs have recently been gaining a considerable amount of interest in the field of biomedicine in terms of drug delivery, protein detection, gene delivery, and DNA purification due to their unique physiochemical properties [63,64], which include high stability and carrier capacity, as well as the possibilities in utilizing various administration routes. NPs are made up of components that are biocompatible which can either be from source that is naturally obtained or synthetically produced [65]. They are generally classified in the range of 1–1000 nm but favorably between the range of 5 and 100 nm. NPs have been used to improve permeability and bioavailability of drugs [66]. In that instance, various NPs such as microemulsion [67], nanoinsulin [68,69], and vesicles [70] have been successfully encapsulated and used for transdermal delivery of insulin. For example, King et al. evaluated the effect of an insulin-encapsulated biphasic lipid vesicle composed of components such as soya phosphatidylcholine, cholesterol, propylene glycol, and compound PDM27 in the lipid phase which was hydrated with microemulsion aqueous phase of linoleamidopropyl-PG-dimoniumchloride phosphate, olive oil, methylparaben, and propylparaben. The oil-in-water emulsion was subsequently incorporated into patches of suitable size. The hypoglycemic effect in rats treated with the biphasic patch was evident up to 77 h, indicating the sustained release of insulin [70]. A follow-up study was carried out to understand the involvement of lymph nodes in transdermal insulin delivery. In this study, however, rats were given varying concentrations of recombinant human insulin at 1 mg, 2 mg, 5 mg, and 10 mg for 73 h. With this, it was seen that the drop in blood glucose level was very much dependent on the dose, as a higher concentration of insulin increases the time and magnitude of response in blood glucose level. Additionally, a steady increase in insulin was observed in the inguinal lymph nodes, where the level of insulin was detected as early as 6 h into the treatment, followed by a constant increase up to 73 h. These results suggest that the accumulation of insulin in the area of the skin where the biphasic patch was administered can act as a reservoir for the controlled release of insulin into the systemic circulation via lymph drainage [71].

In another study, Gold nanorods (GNRs) based insulin (INS) complex in an oil phase were developed to form a solid-in-oil (SO) formulation (SO–INS–GNR). The high permeability of insulin was observed in rats treated with SO-INS-GNR with the addition of near-infrared light (NIR) light irradiation. This is because the Gold nanorods in the complex have the ability to absorb irradiation from the NIR, converting light energy into heat, which eventually breaks the SC layer (Figure 2D) [72]. Similarly, a solid-in-oil (S/O) nanodispersion technique containing insulin and oligo-arginine peptides effectively enhanced the permeation of insulin into the skin. The permeability of insulin was 4.5-fold higher in S/O nanodispersion with R9 peptides as compared to S/O nanodispersion only. Histopathological study indicates the involvement of isopropyl myristate (IPM), a component added along with the formulation of the nanodispersion (Figure 4) and protein transduction domain (PTD) in disrupting the SC layer and improving the penetration of insulin into the skin [25]. Insulin-loaded microemulsion containing 10% oleic acid, 38% aqueous phase, and 50% surfactant phase with 2% DMSO demonstrated an excellent permeation flux of insulin in vitro. The preliminary data suggest a potential use of microemulsion for the transdermal delivery of insulin; provided further analysis on this formulation is evaluated in animals for its efficacy and long-term sustainability [67]. 

Calcium carbonate (CaCO3) nanoparticles have been reported for their successful sustained release of incorporated drugs [68]. Following this, insulin was encapsulated in CaCO_3_ NPs (nanoinsulin) for transdermal delivery of insulin. Significant reductions in blood glucose levels with a slow onset have been reported in mice treated with nanoinsulin compared to subcutaneously administered insulin, indicating differences in insulin absorption. Nevertheless, they found no aggregation or denaturation of insulin using this method [68]. Although this study presented a simple and effective method for transdermal delivery of insulin, the permeability of nanoinsulin through the skin has not been documented, indicating the need for follow-up studies. Similarly, nanoinsulin-loaded chitosan (CS)-polyvinyl alcohol (PVA) (Figure 5) blend hydrogels were developed with the addition of cross-linking agent, glutaraldehyde (Figure 5). The evaluation of the mechanical properties of hydrogel shows a high tensile strength with good elongation at break. This implies good deformability and flexibility in the hydrogel, which could be from the interactions between the polymers used. Additionally, morphological analysis of the hydrogel showed a highly porous film structure which remained unchanged even with the addition of insulin, which indicates a good miscibility in between the hydrogel and nanoinsulin. The porous honeycomb-like structure could have facilitated the permeation of nanoinsulin into the hydrogel, and this was further validated upon the detection of elemental sulfur, which is a crucial element of insulin in the hydrogel. Following this, in vitro drug release of the hydrogel demonstrates a good permeability of nanoinsulin which was in accordance with Fick’s first law of diffusion. This, therefore, concludes the use of nanoinsulin-loaded hydrogels as potential transdermal enhancers for the delivery of insulin [69]. 

Sadhasivan et al. had successfully formulated a nanoencapsulation containing insulin and chitosan along with varying concentrations of polymers, which were eventually loaded into patches [73]. This study showed that patches formulated with polymers such as polyethylene glycol (PEG) and hydroxypropylmethylcellulose (HPMC) could improve insulin penetration through the skin. Furthermore, patches with HPMG 60% and PEG 40% concentrations proved to be the most permeable to insulin compared to other concentrations. The thermodynamic activity of patches was assumed to be due to the presence of functioning glycols along with the combination of solvents in the formulation [73]. Interestingly, another study developed insulin NPs using a new micronization technique called the Super-Critical Antisolvent (SAS) process. This technology was chosen because it is cost-effective and environmentally friendly. The use of the SAS process yielded a uniform spherical nanoparticle without any form of degradation to the encapsulated insulin. In vitro analysis of insulin-NPs showed a high permeation rate in accordance with Fick’s first diffusion law. This indicates the potential use of insulin-NPs for the transdermal delivery of insulin, which warrants further study in animal models [74].

In a recent study, the physiochemical properties of poly [oligo(ethylene glycol) methacrylate]-b-poly[(vinyl benzyl trimethylammonium chloride)] (POEGMA-b-PVBTMAC) diblocks and poly[oligo(ethylene glycol) methacrylate-co-(vinyl benzyl trimethylammonium chloride)] (P(OEGMA-co-VBTMAC)) copolymers of different composition followed by insulin encapsulation were studied. The addition of the copolymer with insulin has been reported to show good stability and protect proteins from adsorption and aggregation. In addition, electrostatic interactions between oppositely charged polyelectrolytes and biopolymers have been identified, leading to the potential use of these polyelectrolyte systems in the transdermal delivery of insulin [75]. These formulated NPs effectively encapsulated and delivered insulin across SC membranes and sustained insulin release, thereby prolonging the decline of blood glucose levels. Moreover, the structural integrity of insulin was maintained without degradation throughout the encapsulation process and after the release of insulin from NPs.

#### 3.1.4. Carrier

Similar to NPs, various delivery systems have been developed and tested for improved insulin permeability through the skin [76]. The use of a flexible lecithin vesicle (Figure 6) as a carrier for insulin was formulated and compared to a conventional vesicle for insulin transdermal activity. The difference between flexible vesicles and conventional vesicles was the presence of sodium cholate. In vivo analysis of insulin-containing flexible vesicles on the abdominal skin of mice showed a significant reduction in blood glucose levels, with a 50% reduction after 18 h. However, no hypoglycemia was observed in rats treated with conventional vesicles, suggesting the importance of sodium cholate in promoting transdermal insulin delivery. This study suggested that the presence of sodium cholate conceives vesicles that are flexible enough to penetrate SC crevices. Furthermore, sodium cholate can potentially alter the physicochemical properties of lipid bilayers by exerting a strong influence on the order of lecithin alkyl chains, ultimately increasing bilayer fluidity and vesicle flexibility (Figure 2E) [21]. 

In another study, the use of transferosome gels (highly deformable vesicles) composed of insulin for transdermal delivery of insulin showed good permeability to pig skin in vitro. Furthermore, an in vivo analysis in diabetic rats revealed sustained hypoglycemia, demonstrating the potential efficacy of transferosomes in the transdermal delivery of insulin [77,78]. On the other hand, arginine-based unsaturated poly(esteramide) (Arg-PEA) hydrogels have been used as carriers for transdermal delivery of insulin due to their improved biocompatibility, biodegradability, and water solubility. In fact, they can also form strong hydrogel matrices by cross-linking with other polymers, such as polyethylene glycol diacrylamide (PEG-DA). This study concludes the efficacy of hydrogels in lowering blood glucose levels in diabetic mice, demonstrating good release of insulin from hydrogels. Furthermore, skin irritation analysis showed that these hydrogels could be safely applied to the skin without irritation [79]. 

Recently, green nano formulations using ginsenosides (GS) (Figure 6) as nanodrug carriers to encapsulate insulin were investigated. GS are natural triterpenoid saponin compounds that could essentially create transient pores in the membrane via interactions with components like phospholipids and steroids, which could promote skin penetration, disruption of intracellular lipid barriers of the SC, and increase the distribution of drugs into the SC. That said, GS has been observed to transiently permeate the skin barrier, which facilitates insulin penetration into cells. Furthermore, the accumulation of insulin GS nanocarriers in the skin enabled controlled and sustained release of insulin for up to 48 h [24]. Moreover, the use of GS inherently protects insulin from hydrolytic enzymatic degradation, representing an excellent alternative for the transdermal delivery of insulin. 

#### 3.1.5. Peptides

A short synthetic cyclic peptide, ACSSSPSKHCG, was identified by in vivo phage display, and this peptide significantly enhanced the transdermal delivery of insulin deep into the hair follicle [80]. Following this, Chang et al. suggested a series of cationic cyclopeptides of TD-1 peptide sequences partially substituted with arginine or lysine to determine the permeation in Caco-2 cells. It was noted that TD-34 substituted lysine in N-5 and N-6 exhibited the best enhancement activity. They also observed that TD-1 peptide might improve the transdermal absorption of insulin by disrupting the follicle of epithelium tissues. In summary, this study concludes the use of Caco-2 cell monolayers (BL → AP) as a potential preliminary method to determine the activity of transdermal peptide enhancers [81].

#### 3.1.6. Virtual Design Algorithm for Screening of Chemical Enhancers

A computational molecular design employing virtual design algorithms combined with quantitative structure-property relationships (QSPR) was used to predict the properties of 43 different chemical permeation enhancer (CPE) functional groups. CPE screening was performed using changes in electrical resistance, an alternative to the traditional labor-intensive method, and from those tested, 22 CPEs were selected for further analysis. This study demonstrated that none of the functioning groups were particularly responsible for the transdermal enhancement of insulin, and instead, CPEs that has low positive log K_ow_ and/or has at least one hydrogen donor or acceptor not limited to toluene are considered good enhancers [82,83]. That said, the use of this virtual design could substantially reduce the time and cost of identifying potential CPE for effective transdermal insulin delivery.

### 3.2. Transdermal Delivery of Other Pharmaceutical Drugs

#### 3.2.1. Dimethyl Sulfoxide 

DMSO has long existing in the pharmaceutical industry dating back to the 19th century. Despite being clinically controversial, the use of DMSO as a solvent for various insoluble drugs is still very much practiced [84]. In addition, DMSO is known for its ability to transport smaller molecules across the skin and mucous membranes [85,86]. This is largely due to its amphiphilic nature, where DMSO interacts with lipids at the stratum corneum, altering the structure of the peptide, which then changes the partition coefficient and therefore increases the permeability of molecules [87]. One of the earliest studies on the efficacy of DMSO in enhancing transdermal permeation of 14C-labeled propan-1-ol and hexan-1-ol was performed on the stratum corneum of neonates. This study was of great importance as it confirms the importance of DMSO concentration in improving permeability. It is suggested that a concentration between 70% and 80% of DMSO increases the permeability of the stratum corneum. However, anything above 80% could lead to irreversible damage to the stratum corneum. A rational explanation for such an effect is due to the hydrogen bonding of DMSO and its interaction with stratum corneum lipids, which increases permeability. However, this can only be observed when free DMSO molecules are present, corresponding to concentrations of 70% or higher [88]. Similarly, the permeation of bisoprolol fumarate was examined using different concentrations of chemical enhancers (Tween 80, propylene glycol, and DMSO). This study also emphasized the importance of choosing the optimal concentration of permeation enhancer. It is reasonable to say that higher concentrations of enhancers do not necessarily lead to higher fluxes of molecules across the skin [89]. 

Another study compared the efficacy of a topical solution of diclofenac sodium (TDiclo) and DMSO with the use of diclofenac, an oral non-steroidal anti-inflammatory drug (NSAID), in treating knee osteoarthritis. Topical administration of TDiclo is as effective as low-dose oral NSAIDs, with less absorption into the systemic circulation. Additionally, using TDiclo reduces the side effects typically associated with oral diclofenac. The most frequently observed side effect of TDiclo was dry skin which could be in relation to the use of DMSO as a vehicle. However, the use of emollients after treatment may be prescribed in order to alleviate the dryness of the skin [90]. The use of Aloe vera as a permeation enhancer in the delivery of lidocaine was compared with the use of DMSO. At the highest concentration (3%), Aloe vera was able to increase lidocaine penetration by 79.18% compared to 84.52% for DMSO. From this, we can conclude that DMSO is still a potent enhancer and can be used for future screening of various chemical enhancers [91]. A recent study incorporating DMSO into a transdermal patch for the delivery of estradiol was evaluated [92]. This study concluded the benefits of using DMSO in the formulation of self-adhesive (DIA) patches compared to conventional patch formulations. One of them is certainly the improved permeability of estradiol. Furthermore, the use of DMSO greatly inhibits the recrystallization of estradiol, indirectly reducing skin permeability. However, further studies need to be performed to better determine the most efficient concentrations of DMSO and estradiol in transdermal patch formulations [92]. Idoxuridine is another example of a topical drug that has been formulated with DMSO as a permeation enhancer. The use of 5 or 40% idoxuridine in DMSO has been shown to accelerate healing and shorten pain in focal herpes simplex virus (HSV) and herpes labialis infections. Whereas 30% idoxuridine in DMSO has been shown to shorten the duration of viral shedding in recurrent and primary human genital HSV infections [93]. 

#### 3.2.2. Ionic Liquids 

ILs have been used for the study of various APIs due to several advantages (Section 3.1.1). That said, a recent study was conducted to understand the anti-microbial activity of hydrophobic [BMIM][PF6] and hydrophilic [HMIM][CL] ILs. Despite both exhibiting over 5% anti-microbial activity, hydrophobic ILs penetrate deeper into skin layers than hydrophilic ILs [94]. This is because hydrophobic ILs promote the utilization of the transcellular pathway within the subcutaneous layer where the epithelial membrane is disrupted [95]. In another study, the mechanistic fundamental of CAGE, an IL at a ratio of 1:2, were investigated with dextran as a model drug. Interestingly, dextran with a molecular weight of 150 kDa was solvated in CAGE and exhibited transdermal capabilities via inducing lipid extraction. The extracted lipid would then be replaced with IL and water which allows a quick diffusion of the drug [96]. On another hand, IL-assisted transdermal delivery systems for a sparingly soluble drug such as acyclovir were investigated. Acyclovir is an anti-viral drug used for treating genital herpes, varicella-zoster virus, Epstein–Barr virus (EBV), and cytomegalovirus (CMV) [97]. Though effective, acyclovir given orally has poor bioavailability, whereas topical cream has shown limited effectiveness. Given that the high solvability of acyclovir was seen in IL- dimethylimidazolium dimethylphosphate [C1mim][(MeO)_2_PO_2_], it was chosen out of the other tested ILs. Subsequently, the permeability of acyclovir was observed using [C1mim][(MeO)_2_PO_2_] droplets formulated in isopropyl myristate using a surfactant, polyoxyethylene sorbitan monooleate (Tween-80), and a cosurfactant, sorbitan laurate (Span-20). An increase in transdermal permeation of acyclovir was observed when used with the said microemulsion system as a drug carrier [98]. 

In contrast to commonly used ILs, recent studies suggest the use of highly biodegradable and less toxic components such as amino acids for the production of ILs. A variety of amino-acid ester-based IL (AAE)Cl was screened using hydrocortisone and 5-Fluorouracil as a model drug. Amongst those tested, three (AAE)Cl were picked, namely glycine methyl ester hydrochloride [GlyC1]Cl, L-proline methyl ester hydrochloride [LProC1]Cl, and L-leucine methyl ester hydrochloride [L-LeuC1]Cl. The ester sites of the different carbon chain (eight and twelve) in each ILs were altered in order to increase permeability. Among these, [LProC12]Cl and [L-LeuC12]Cl demonstrated enhanced penetrating activity by interacting with the intercellular lipid domain by lipid fluidization and extraction [99]. This corroborates with hypothesis suggested by Janusiva et al., where amino acid that has a hydrophobic tail attached to an amino acid head via linkages made out of an ester bond could be of use in intercalating and disrupting the stratum corneum lipid arrangement [100,101]. In another study, fatty acid-based amino acid IL (FAAAE-IL) was proposed as a new eco-friendly solvent for transdermal drug delivery systems. The aim of this study was to discover approved pharmaceutical ingredients that are biocompatible and have low skin toxicity. It was concluded that FAAAE-IL can enhance the penetration of ibuprofen and peptides into the skin by fluidizing lipids of the stratum corneum [102]. The intercellular lipid mobilization interaction by permeation enhancers can also be explained as dependent on the hydrocarbon regions of the lipid bilayer. Improved fluidity leads to the effective flux of molecules. In addition, it is also mentioned that the n-alkyl group chain was able to induce membrane permeation and intercalate into the lipid bilayer to increase fluidization of the lipid bilayer, while the polar head group locates in the polar or semipolar microenvironment. It is also worth noting that the polar heart group does not provide potentiation activity but only assists in the movement of enhancers to their sites of action by free energy transfer from the primary aqueous phase to the semipolar microenvironment of lipid structures [103,104,105]. From the same perspective, enhancers should at least consist of a polar head and a hydrophobic chain to ensure interaction with ceramide for efficient augmentation activity. Ceramides are essential molecules that ensure a tight lamellar lipid organization with resistance to the external environment [106]. 

#### 3.2.3. Deep Eutectic Solvents 

DES is a new-generation solvent that has recently gained ample recognition in the pharmaceutical field. They are a mixture of two or more compounds at a specific ratio that achieves a low melting point. DESs are comparably low-cost and effective as an alternative to ILs or organic solvents [107]. Additionally, DESs are arguably known to be thermally stable, tunable, less volatile, low vapor pressure, less toxic, and highly biodegradable. As of recent, a few classes of DESs have been identified, which include natural deep eutectic solvents, as the name suggests, components used for the synthesis of natural deep eutectic solvents are obtained from natural sources such as sugar, amino acids, and organic acids [108,109]. Therapeutic deep eutectic solvents are eutectic mixtures containing APIs as part of the formulation [110]. DES possesses excellent properties that complement some of the obstacles of existing solvents, especially in the area of drug delivery. Conjugation of APIs to DES enables a variety of drug delivery routes, including transdermal delivery. 

The most extensively researched area of DES-based drug delivery systems is therapeutic carriers. In a study, the use of a eutectic system comprised of capric acid and menthol was evaluated for the solubility of fluconazole and mometasone furoate for the possibility of transdermal application. The use of DES increased drug solubility, possibly due to the formation of hydrogen bonds between capric acid and the drug. Furthermore, a topical cream composed of both fluconazole and mometasone furoate showed no signs of skin irritation, edema, or inflammation when applied to albino rats [111]. Similarly, in another study, ibuprofen, along with various terpenes, was prepared into a binary eutectic mixture which exhibited an increase in transdermal delivery across the abdominal cavity of Caucasian humans. It was also observed that ibuprofen–eutectic mixture showed a much higher flux as compared to when ibuprofen is delivered alongside just a singular component of DES, suggesting the involvement of melting point depression [112]. This is in lieu of another reported studies where an inverse correlation between melting point and transdermal capability was noted [113]. Kang et al. also reported the permeability of lidocaine across snake skin in a concentration-dependent manner using a eutectic mixture containing menthol [114]. Likewise, the permeability of testosterone across human cadaver skin was observed with a eutectic mixture of menthol [115]. The use of menthol as a part of the eutectic mixture allowed the alteration of skin lipids, further enhancing the movement of drugs across the stratum corneum. 

In another study, APIs-DES transdermal patch was developed using rotigotine for the treatment of Parkinson’s disease [116]. The transdermal patch was optimized and dispersed in hydroxyl pressure-sensitive adhesive, which was seen to exhibit a significant amount of skin permeation. It also has good drug-polymer miscibility, which further enhances transdermal drug delivery. In addition, in vivo pharmacokinetic studies show efficacy comparable to the commercial product [116]. An ion-gel system of DES (choline chloride-ascorbic acid) and 2-hydroxyethyl methacrylate was developed as a carrier for the delivery of the anti-cancer drug sunitinib. Drug release was reported to occur much more rapidly at a low pH of 1.2 compared to the high pH of 6.8 and 7.4, leading to the conclusion that drug release from ion gel is indeed pH dependent. [117]. On the other hand, Lodzki et al. successfully created cannabidiol ethosome using a eutectic mixture consisting of cannabidiol and phosphotidylcholine. The eutectic system aided the permeability of cannabidiol across the skin membrane, which is otherwise not permeable with low bioavailability [118]. 

A non-invasive transdermal strategy using mesoporous silica NPs (MSN) and DES (amino acids and citric acid) showed significant improvement in skin permeability. MSN-DES was prepared by using citric acid to modify the surface of MSN, followed by heating the amino acid lysine. The mechanistic action of the MSNs-DESs was associated with the ‘erosioning’ effect of the stratum corneum whereby the tight structure of the stratum corneum is altered, allowing fluidization of membrane, hence enabling the permeation of MSNs deep into the skin. Despite this, no evidece of tissue necrosis or edema was observed in the tested animals [119]. Another study used a combination of DESs (4-methoxy salicylic acid-betaine, lactic acid-betaine, malic acid-betaine, malic acid-choline chloride, malic acid-l-carnitine, mandelic acid-betaine, phytic acid-betaine) in skin permeability tests. This study highlights the importance of the concentration and viscosity of DESs for transdermal release. As an example, 1000 mg/mL of phytic acid-betaine (3:1) aqueous solution showed the highest permeation as compared to other concentrations or viscosity of the DESs mixture. Furthermore, it is also reported that individual component betaine had lower permeability as compared to the use of DESs-betaine, signifying the importance of hydrogen bond formation to improve betaine permeability [120]. On the other hand, pharmaceutical DESs (PDESs) (catechol: ChCl (1:1), imipramine HCl: glycerol (1:2) and ascorbic acid: ChCl (1:2)) and gelatin were evaluated to understand the transdermal flux rate of APIs. An increase in transdermal permeability was observed using PDES, demonstrating the importance of DES formation in transdermal drug delivery [121]. A recent study investigated carboxylic acid terpene-based DES. This study reported that the use of L-menthol-based DES improved the penetration of resveratrol. This is because they have long hydrocarbon chains that are better at breaking down lipid membranes compared to short hydrocarbon chains. Additionally, thymol-based DESs were able to extract lipids in the stratum corneum providing better permeation to the corneocytes. Overall, DESs with longer hydrocarbon chains could be one of the key criteria in selecting permeation enhancers. [122]. The versatility of DESs displays great opportunities in the field of drug research and a detailed perspective on the application of DESs in the said field has been discussed [123].

#### 3.2.4. Essential Oils

Essential oils are obtained from the extraction of secondary metabolites from plants or natural products, which are mixtures of different aromatic compounds [124]. The uses of essential oils are extremely diverse, with a range of essential bioactivities, including anti-fungal, anti-bacterial, anti-cancer, anti-oxidant, and promotion of wound healing [125]. They are also used as preservatives in the food industry and can be used as an alternative to synthetic pesticides [126]. Over the years, essential oils have been investigated for use as permeation enhancers to deliver essential drugs such as ibuprofen [127], ketoconazole [128], estradiol, 5-fluorouracil (5-FU) [129], p-aminobenzoic acid [130], and labetalol hydrochloride. Although the detailed action of essential oil is not well understood, several studies have shown several potential mechanisms for the flux of drugs across the skin membrane. The essential oil tulsi (Ocimum sanctum) and turpentine showed potentiating abilities, but turpentine showed the best and most effective drug flux. The effect of turpentine oil on microscopic skin showed disruption of the normal stratification of the corneal layers, whereas tulsi oil resulted in extensive destruction of the corneal layer accompanied by condensation of the normal stratified corneal layers [131]. Another study using basil (Ocimum basilicum) accelerated penetration of the lipophilic indomethacin but less so with the hydrophobic 5-FU. The penetration mechanism may be related to the effective distribution of essential oil between the stratum corneum. This reduces the polarity of the stratum corneum and facilitates penetration of lipophilic indomethacin into the skin [132]. Similarly, another study found that basil essential oil effectively delivered the hydrophilic drug labetalol hydrochloride by interacting with the intracellular lipids of the stratum corneum [133]. On the other hand, Alpinia oxyphylla oil also shows a higher affinity for lipophilic corneal horn by reducing the polarity of the corneal horn and allowing lipophilic indomethacin to cross the skin. It was found that the skin penetration activity of Alpinia oxyphylla was contributed by the increase of skin excipients. This study also highlighted that changes in transepidermal water loss were negligible in most cases, with no irritation or toxicity [134]. 

It has been suggested that the presence of terpenes in essential oils increases their interaction with ceramides in the stratum corneum via hydrogen bonding, resulting in the loss of tight lamellar organization for drug penetration [135]. This is observed in terpenes with more alcohol groups compared to terpenes with a carbonyl group. In contrast, the enhancement capability of camphor was significantly higher than that of geraniol, thymol, and clove oil despite the presence of more alcoholic oxygen atoms in these oils. In this case, camphor is said to have a lower boiling, hence being less self-associated, allowing a more free interaction with lesser energy towards the lipid at the stratum corneum. Comparatively, geraniol, thymol, and clove oil demonstrate higher self-association, meaning more energy is needed to interact with the lipid bilayer. The use of terpenes as penetration enhancers increases competing hydrogen bonds with lipids, disrupting the bilayer and lowering the activation energy of drugs or molecules that penetrate the skin. [133]. Studies on Magnolia fargesii show that both theophylline and cyanidanol are effectively delivered through the skin. Additionally, Magnolia fargesii essential oil has been tested for safety, and the results show that Magnolia fargesii is safe for use as a chemical penetration enhancer [136]. Vashisth et al. showned that all essential oil tested (aloe vera, tea tree oil, cumin oil, and rose oil) had penetrating capacity for the tested drug losartan potassium, but only aloe vera oil demonstrated 2.36 fold enhancement of losartan potassium flux over control (without enhancer) [137]. The mechanism of action of aloe vera oil in the release of losartan potassium is related to the disruption of hydrogen bonds of the drug with intracellular lipids, suggesting a novel polar pathway in which oil interacts with the polar head group region of the stratum corneum lipid bilayer [137]. Overall, essential oils exert their skin penetration capabilities by altering the structure of the stratum corneum barrier and interacting with intracellular lipids to diffuse drugs throughout the skin. In addition, they also exhibit skin-vehicle separation properties, allowing drugs to readily migrate across skin membranes [126].

In addition, essential oil-derived compounds such as terpenes also play an important role as penetration enhancers [138]. For instance, limonene was shown to increase the permeability of ketoprofen, which was 3-fold over controls [139]. In another study, the use of limonene for the delivery of aceclofenac was compared against a widely used non-ionic surfactant, Span-20. The results show a significant increase in drug permeability with limonene compared to Span-20, with further increases in drug in-flux as the enhancer concentration increases. Furthermore, aceclofenac released from limonene-containing transdermal patches showed significant anti-inflammatory effects [140]. Similarly, the transdermal efficacy of limonene combined with co-surfactants (ethanol, isopropanol, and propylene glycol) as a microemulsion showed great potential in delivering curcumin in neonate pig skin as compared to other enhancers (1,8-cineole and α-terpineol) [141]. Propranolol skin permeation using polymer films containing cineol showed higher drug permeation compared to menthol and propylene glycol alone. A mixture of both cineol and propylene glycol also showed a significant increase in skin permeation comparable to that of the individual cineoles. Furthermore, it is suggested that the composition of the polymeric film would be ethyl cellulose: polyvinyl pyrrolidone: propranolol hydrochloride at 6:3:4 and 10% (*w*/*w*) cineole and 7:2:4 and 10% (*w*/*w*) propylene glycol and cineole, respectively [142]. The effect of terpenes (menthol, cineole, terpineol, menthone, pulegone, and carvone) in delivering imipramine hydrochloride across rat skin was evaluated. Amongst tested terpenes, menthol exhibits the highest flux of imipramine hydrochloride, comparable to that of cineole as well. This is attributed to the breaking of hydrogen bonds between ceramide heads by terpenes either by donating or accepting hydrogen bonds or through weak self-association [135]. 

### 3.3. Section Summary

Since insulin injections are the only option available to patients with T1DM, the transdermal route of administration offers an excellent alternative to traditional insulin delivery methods. Various possible techniques have been extensively evaluated for the transdermal delivery of insulin, with significant effort given toward the use of chemical enhancers. A number of studies have also revealed possible mechanisms involving the use of chemical enhancers for insulin permeation (as discussed earlier; Table 2). More importantly, there was no evidence of irreversible damage or alterations to the skin, suggesting that the use of chemical enhancers for transdermal insulin delivery is indeed safe and feasible. 

## 4. Concluding Remark and Future Outlook

Overall, transdermal drug delivery approaches offer significant advantages over traditionally used oral drugs. This means addressing important issues such as low bioavailability, high side effects, and pre-systemic metabolism. However, the stratum corneum forms a rigid barrier that prevents the influx of drugs into the body. Extensive research efforts have therefore been undertaken to recognize the potential of enhancers (chemical and physical) for drug delivery. In most studies, selected chemical enhancers were able to support diffusion of insulin and other therapeutic agents across the skin, and some even exerted biological functions. Possible mechanisms of drug penetration include reversible disruption of the stratum corneum, resulting in increased drug permeability. Enhancers may also facilitate drug partitioning by altering stratum corneum properties by interacting with the intercellular protein domains. Furthermore, unsaturated fatty acids can be used to the increase fluidity of the lipid domain for the transverse of drugs. This is largely due to their kinked chain that leads to higher lipid disruption.

Despite great success, the use of chemical enhancers is still very much in its infancy despite being proven effective in both in vivo and ex vivo models. In fact, none of these enhancers have made it to the clinical setting. There are still many developmental limitations that need to be optimized before being used for the transdermal delivery of drugs. Firstly, most penetration studies have been performed in rodents, but these results may not be applicable to humans, as skin composition varies widely between species. The main difference between human skin and rodent skin is the composition of the skin layers and their functions. Basically, both humans and mice have three specialized layers (epidermal, dermal, and subcutaneous layer). Because rodents have relatively thin epidermis and dermis, loosely attached skin, and lack of sweat glands, with the exception to the paws, drugs permeate through the skin more easily in rodents than in humans. Secondly, the short- and long-term safety of the use of these enhancers is limited, opening various possibilities of adverse side effects. In that sense, the likelihood of risk associated with skin irritation, swelling, edema, and damage should also be evaluated. In addition to that, a higher concentration of chemical enhancers was associated with an increase in the permeability of drugs, but this could induce possible skin irritation. 

On the other hand, there are also reports that the amount of insulin is higher if administered with enhancers compared to subcutaneous injections. The rationale is that higher concentrations may facilitate passive diffusion across the skin barrier, but the downside to this is that insulin has little margin when it comes to safety, and insulin overdose may result in severe symptoms, including hypoglycemic coma, neurological impairment, and even death. Hence, this warrants further in-depth study on the bioavailability of insulin before clinical application. Additionally, several steps may be taken to carefully select enhancers that can deliver insulin in a designated therapeutic dose. In this case, the screening of potential enhancers and their synergistic mixture of chemical enhancers could be done using a computerized mechanism for rapid and cost-reduction development with the use of in-silico modelling such as the QSPR model. Nevertheless, different formulations of chemical enhancers have different synergistic effects under different formulation conditions, which is why, not all enhancers can be compared to those used for the delivery of insulin. Additionally, many of the enhancers used may not be suitable for insulin delivery due to their denaturing properties and should be evaluated prior to insulin use. Along with that, it is also worthy to note that despite the various advancement in using DESs as a transdermal carrier, they have not been implicated for the transdermal delivery of insulin; however, a study on the alternative route (nasal) for insulin delivery using DESs have been carried out. This suggests that DESs could potentially occupy the role in the transdermal delivery of insulin which should be highly considered.

Most of these studies have had great success with transdermal insulin delivery, but the exact mechanism of insulin permeation through the skin is still poorly understood. At the same time, the techniques used to determine the efficiency of chemical enhancers in insulin delivery vary from one study to another (from virtual screening to cell lines or animal models), leading to gaps in safety and consistency issues. Ideally, selected enhancers should be pharmacologically inert, or at least, known for their action against the stratum corneum to ensure that the modification of the skin is reversible in long-term use. They should also be safe, non-irritant, non-toxic, and able to give a sustainable effect for a prolonged period. Importantly, these enhancers must not chemically or physically alter the function or structure of insulin, which could impair its efficacy. Therefore, a thorough evaluation of these reported chemical enhancers should be performed in the context of various other established insulin delivery studies. We believe that if these issues are resolved, the use of chemical enhancers for transdermal insulin delivery in humans will be expanded to allow detailed pharmacokinetic and pharmacodynamic studies to be conducted through clinical trials.

## Figures and Tables

**Figure 1 biomedicines-11-00664-f001:**
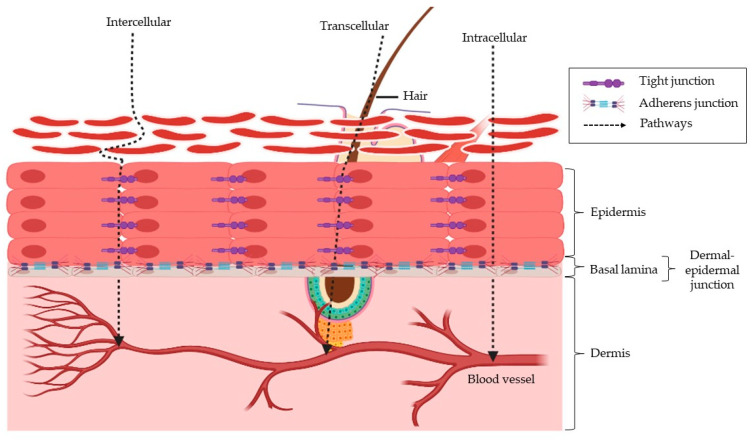
Anatomy of the skin layers and the different pathways (intercellular, transcellular and intracellular) that could be utilized for transdermal drug delivery.

**Figure 2 biomedicines-11-00664-f002:**
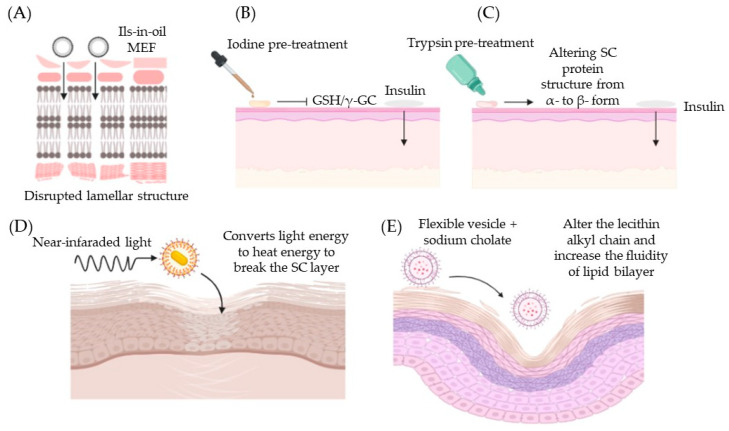
Examples of different types of chemical enhancers and the mechanism of action for the transdermal delivery of insulin. (**A**) Disruption in the lamellar structure of the stratum corneum (SC) was seen with the use of ILs in oil microemulsion (MEF), allowing the permeation of insulin into the skin. (**B**) Pre-treatment of iodine inhibits sulfhydryl compounds such as glutathione (GSH) and gamma-glutamylcysteine (γ-GC) which then aids the permeation of insulin. (**C**) Pre-treatment with trypsin allows the alteration in the protein structure of the SC from the alpha- to the beta- form allowing the permeation of insulin. (**D**) Gold nanorods (GNRs) based insulin (INS) complex in an oil phase (SO) with the addition of near-infrared light was administered. The complex was then able to absorb light energy to heat energy, breaking the SC layers and allowing the permeation of insulin to the skin. (**E**) A flexible vesicle with the addition of sodium cholate alters the lecithin alkyl chain and increases the fluidity of the lipid bilayer easing the vesicle into the skin.

**Figure 3 biomedicines-11-00664-f003:**
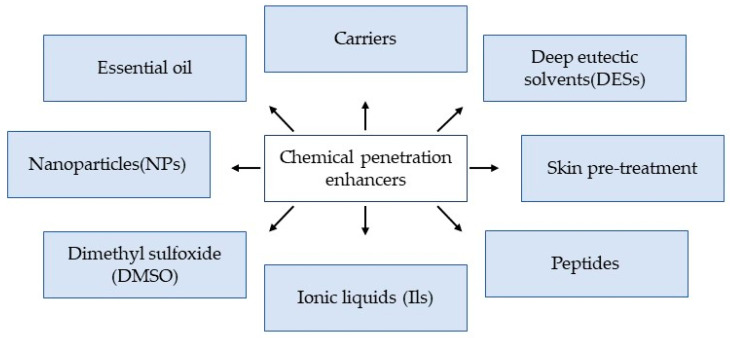
Different types of chemical enhancers that have been used for the delivery of insulin and other pharmaceutical drugs.

**Figure 4 biomedicines-11-00664-f004:**
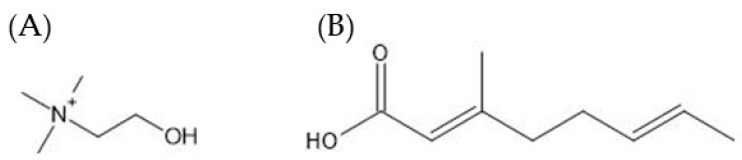
Chemical structure of individual components of CAGE, (**A**) choline bicarbonate and (**B**) geranic acid.

**Figure 5 biomedicines-11-00664-f005:**
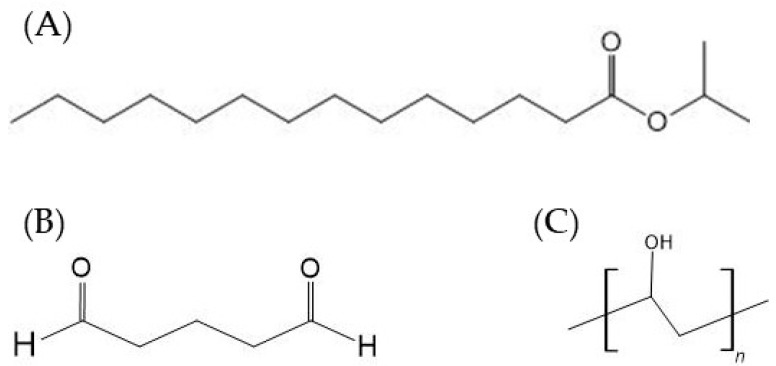
Chemical structure of (**A**) Isopropyl myristate, (**B**) glutaraldehyde, and (**C**) polyvinyl alcohol.

**Figure 6 biomedicines-11-00664-f006:**
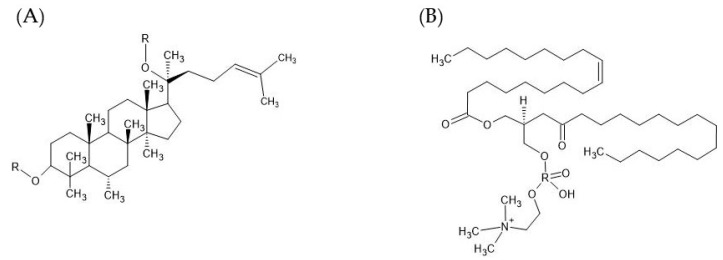
Chemical structure of (**A**) ginsenoside and (**B**) lecithin.

**Table 1 biomedicines-11-00664-t001:** Advantages and disadvantages of transdermal delivery of insulin as compared to other routes of administration.

Routes of Administration	Oral	* IV	Nasal	Transdermal
Avoid first-pass effect	No	Yes	Yes	Yes
Constant drug level	No	Yes	No	Yes
Self-administration	Yes	No	Yes	Yes
Unrestricted activity	Yes	No	Yes	Yes
Better patient compliance	Yes	No	Yes	Yes
Absorption issue	Yes	No	Yes	No

* Intravenous (IV).

**Table 2 biomedicines-11-00664-t002:** Examples of chemical penetration enhancers used for the delivery of insulin and other pharmaceutical drugs, with the possible mechanism of action summarized.

Chemical Penetration Enhancers		Pharmaceutical Drugs	Transdermal Permeation Model	Mechanism of Action	References
ILs	Choline bicarbonate and geranic acid (CAGE)- ratio 1:2	Insulin	Ex vivo (Porcine skin) In vivo (Diabetic male Wistar rats)	Extracting lipids from the stratum corneum	[56]
	(CAGE)- ratio 1:2, 1:4	Insulin	Ex vivo (Porcine skin)	Extracting lipids from the stratum corneum	[22]
	Microemulsion of choline-fatty acids [Chl][FAs]	Insulin	Ex vivo (Yucatan micro-pig skin) In vivo (Diabetic mice)	Activating the fluidizing effect on the lipid bilayer, altering the lamellar structure of the SC lipid	[57]
Skin pre-treatment chemicals	Iodine (povidone-iodine ointment)	Insulin	In vivo (Diabetic Sprague–Dawley rats)	Inactivate thiols on the skin which would significantly increase the bioavailability of active insulin	[23]
	Trypsin	Insulin	Ex vivo (Male Wistar rat skin) In vivo (Diabetic male Wistar rats)	Alter the stratum corneum structure from alpha- to beta-form and decreases the electrical resistance of the skin	[60]
NPs	Biphasic insulin vesicles patches	Insulin	In vivo (Diabetic Sprague–Dawley rats)	-	[70]
	Solid-in-oil Gold nanorods	Insulin	Ex vivo (Sea:ddY male mice skin) In vivo (diabetic Sea:ddY male mice)	Gold nanorods absorb irradiation from near-infrared light converting light energy to heat energy which eventually breaks the stratum corneum layer	[72]
	Solid-in-oil nanodispersion containing oligo-arginine peptides	Insulin	Ex vivo (Yucatan micro-pig skin)	Disruption to the stratum corneum layer	[25]
	Calcium carbonate (CaCO3)	Insulin	In vivo (Normal ddY mice, diabetic dB/dB, and kkAY mice)	-	[68]
	Water-in-oil nanoemulsion containing oleic acid	Insulin	Strat-M^®^-surrogate to human skin	-	[143]
Carriers	Flexible lecithin vesicles with sodium cholate	Insulin	In vivo (Normal Kunming mice)	Sodium cholate could alter properties of lipid bilayer (lecithin alky chain) to increase fluidization of and flexibility of vesicle	[21]
	Transferosome gel	Insulin	Ex vivo (Goat skin) In vivo (Diabetic Wistar rat)	Membrane fluidization and stratum corneum alteration	[77,78]
	Arginine-based unsaturated poly (ester amides) (Arg-PEA) based hydrogel	Insulin	In vivo (Diabetic ICR mice)	-	[79]
	Ginsenosides based nano-formulation	Insulin	Ex vivo (Sprague Dawley rat skin)	Ginsenosides are natural triterpenoid saponin compounds that could essentially create transient pores in the membrane via interactions with components like phospholipids and steroids, which could promote skin penetration, disruption of intracellular lipid barriers of the stratum corneum	[24]
DMSO		colloidal carrier systems consisting of DMSO	Ex vivo (Male Wistar rat skin) In vivo (Male Wistar rat skin	-	[144]
		Lecithin/isopropyl myristate reverse micelles	Ex vivo (Hairless mice) In vivo (Diabetic New Zealand white rabbits)	-	[145]
		^14^C labeled propan-l-ol and hexan-l-ol	Ex vivo (Rat skin)	Hydrogen bond mediated transdermal permeation	[88]
		bisoprolol fumarate	Ex vivo (Rabbit skin)	-	[89]
		Diclofenac sodium solution	Human skin	-	[90]
		Lidocaine	Ex vivo (dialysis membrane)	-	[91]
		Esterdiol	Ex vivo (Porcine ear skin)	-	[92]
		Idoxuridin	Clinical trial (human skin)		[93]
ILs	Hydrophobic and hydrophilic ILs	[HMIM] [Cl], [BMIM] [PF6]	Ex vivo (Porcine ear skin)	-	[94]
	CAGE—ratio 1:2	Dextran	Ex vivo (female Yorkshire pigs)	Lipid extraction	[96]
	IL- dimethylimidazolium dimethylphosphate [C1mim][(MeO)_2_PO_2_]	Acyclovir	Ex vivo (Yucatan micropig skin)	-	[98]
	Amino-acid ester-based IL (AAE)Cl	Hydrocortisone, 5 -Fluorouracil	Ex vivo (Kunming mice) In vivo (Wistar rats)	Interacting with the intercellular lipid domain by lipid fluidization and lipid extraction	[99,100,101]
	fatty acid-based amino acid ILs (FAAAE-IL)	Ibuprofen, peptide	Ex vivo (Yucatan micropig skin)	Enhancing penetration of drugs across the skin via the fluidizing lipid of the stratum corneum	[102]
	CAGE- ratio 1:2	Curcumin	Ex vivo (Porcine ear skin)	Increase in solubilization of curcumin by the ILs and by momentary changes in the corneal extract.	[146]
DESs	Terpenes (l-Menthol, LD-Menthol, Thymolm 1,8-Cineole)	Ibuprofen	Ex vivo (Human epidermal membranes)	-	[112]
	Lidocaine-L-Menthol	Lidocaine	Ex vivo (Snake skin)	-	[114]
	Menthol-testosterone	Testosterone	Ex vivo (Silastic membrane and nude mouse Skin)	Alteration of skin lipids. Interaction between menthol and testosterone where it increases the solubility of drug, allowing the increase in skin permeation.	[115]
	Rotigotine-Lauric acid	Rotigotine	Ex vivo (Male Wistar rat skin)	Modification of physiochemical properties of drugs.	[116]
	Cannabidiol-phosphatidylcholine ethosomes	Cannabidiol	Ex vivo (CDI nude mice)	Enhanced fluidity of ethosomes through enhancer-membrane and drug-enhancer interactions.	[118]
	Citric acid: Lysine 1:1	Mesoporous silica NPs	Ex vivo (Porcine ear skin, hairless mice skin, biomimetic membrane, rat skin) In vivo (Female hairless mice)	‘Erosion’ effect of stratum corneum whereby the tight structure of the stratum corneum is altered, allowing fluidization of membrane, hence enabling the permeation of MSNs deep into the skin	[119]
	Phytic acid-betaine (3:1)	Betaine	Ex vivo (polymethylsiloxane, mice skin) In vivo (Healthy human skin)	-	[120]
	catechol: ChCl (1:1), imipramine HCl: glycerol (1:2), and ascorbic acid: ChCl (1:2))	-	Ex vivo (Porcine loin cut, saline-soaked bovine hides)	-	[121]
Essential oil	Turpentine oil, tulsi oil (*Ocimum sanctum*)	Flurbiprofen	In vivo (Albino rats)	Turpentine oil exhibited the disruption of normal stratification of the stratum corneum. Tulsi oil, there was an extensive disruption of the stratum corneum with condensation of the normal stratified corneal layers with an increase in epidermal thickness	[131]
	Sweet basil (*Ocimum basilicum*)	Indomethacin	Ex vivo (Wistar rat skin)	Effective partitioning of OB essential oil between the stratum corneum, which in turn reduces the polarity of the stratum corneum, facilitating the permeation of lipophilic indomethacin into the skin	[132]
	Basil oil	Labetalol hydrochloride	Ex vivo (Wistar rat skin)	Interacting with the intracellular lipid of the stratum corneum	[133]
	*Alpinia oxyphylla* oil	Indomethacin	Ex vivo (Wistar rat skin) In vivo (Wistar rat skin)	Reduces the polarity of the stratum corneum allowing the flux of lipophilic indomethacin across the skin.	[134]
	*Magnolia fargesii*	Theophylline and cianidanol	Ex vivo (Wistar rat skin), In vivo (Wistar rat skin)	skin-vehicle partitioning by *Magnolia fargesii*	[136]
	Aloe vera oil	losartan potassium	Ex vivo (Wistar rat skin),	Disruption of hydrogen bonding of drug with intracellular lipid. Low activation energy of losartan potassium with the use of aloe vera oil suggests a new polar pathway where aloe vera oil interacts with the polar head group region of stratum corneum lipid bilayer	[137]
	Limonene oil microemulsion	Curcumin	Ex vivo (Pig ear skin)	-	[141]

## Data Availability

Not applicable.

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
