# Peer review of "Current Development of Chemical Penetration Enhancers for Transdermal Insulin Delivery"

_biomedicines, 2023, doi:10.3390/biomedicines11030664_

Round 1

Reviewer 1 Report (Previous Reviewer 3)

Dear Authors,

I believe that the manuscript is suitable for publication, you can find attached my suggestions, I found some formal mistakes that you can correct easily

Author Response

We would like to thank reviewer 1 for the detailed review on the manuscript. We have amended the manuscript accordingly.

  1. All the previously bolded Figure and Table cited in-text have been removed (Figure 1-6; Table 1 & 2).
  2. As per suggestion, we used the software (ADCLab) to re-draw the chemical structures for Figure 4,5 and 6. 
  3. Reference to row 128 has been added accordingly. 
  4. Reference to row 223 has been added accordingly.
  5. Reference to row 332, 476, 583, 677 have been added accordingly.
  6. Row 515-545, style has been justified.
  7. We have thoroughly check the references. DOI (for References 12, 16, 18, 19, 22, 29, 32, 76, 82, 88, 89) and Journal names for (References 59, 60, 77) have been added. Reference 84 style corrected. Other references were checked. Kindly note that there are book references which have no DOI.
  8. Table 2 Gold nanorod corrected.
  9. We have thoroughly checked the whole manuscript and corrected grammar/typing errors, including "()".  In addition, we have improved the clarity of sentences that were too long.
  10. We appreciate the kind suggestions and efforts of Reviewer 1 to further enhance the quality of the manuscript.

Reviewer 2 Report (Previous Reviewer 1)

Advances in the development of permeation enhancers for the transdermal delivery of insulin is described. In addition, the possible mechanism of action and the potential outlook on the proposed permeation approaches are discussed. The manuscript has been significantly improved and I recommend it for publication. 

Author Response

Thank you for your thoughtful comments and efforts toward improving our manuscript. 

This manuscript is a resubmission of an earlier submission. The following is a list of the peer review reports and author responses from that submission.

Round 1

Reviewer 1 Report

The authors discussed progress in the development of chemical enhancers for transdermal insulin delivery. The influence of ionic liquids and peptides on the percutaneous penetration on drugs was described. The authors also discussed the use of computer-assisted molecular design the virtual design algorithm in combination with quantitative structure-property relationship (QSPR) to predict the properties different functional groups as a chemical permeation enhancer. Although there are several excellent publications in this research area, this is a significant addition to the literature, and I recommend the manuscript for publication.

Author Response

We would like to thank reviewer 1 for the kind comments.

Reviewer 2 Report

This review does not add anything valuable to the many reviews in books and journals already available on the general topics of transdermal delivery and penetration enhancement related to proteins including insulin. Indeed there are many better written and more critically insightful articles available.

The introduction (sections 1 and 2) are very general on the broad topics of insulin therapeutics, transdermal advantages, skin permeation/structure. Section 3 is the main body dealing with enhancers but provides nothing new in terms of information or insight. The conclusion is very superficial and again, demonstrates no in depth analysis of the topic.

Author Response

We thank Reviewer 2 for the critical comments. In this revised version, we have amended the manuscript in accordance to the comments given by other reviewers as well. Just to highlight, this review focuses mainly on the non-invasive method of transdermal delivery of insulin using chemical enhancers. One of the recent developments such as the use of computer-assisted molecular design the virtual design algorithm in combination with quantitative structure-property relationship (QSPR) to predict the properties different functional groups as a chemical permeation enhancer has been discussed. In addition, we have added a few useful insights on the use of chemical enhancers and its possible interaction and function in the permeation of insulin. Moreover, some points were rearranged and more images were added to give better clarity to the manuscript.  We also corrected the English and improved the flow of the manuscript (including conclusion) with the assistance of two senior co-authors who are expert in this field.

Reviewer 3 Report

Dear authors, I believe that manuscript needs major revision that you can find in the attached file.

Author Response

We thank Reviewer 3 for the constructive comments. We have amended the manuscript according to your kind comments, please find the point-to-point response below:.

  1. A brief sentence has been added with regards to the use of chemical enhancers in the delivery of insulin (line 121-130. Page 4). Additionally figure 2 have been moved before paragraph 3.1 (line 131. Page 4)
  2. Chemical structure for Choline bicarbonate and geranic acid has been added as Figure 3 (line 154. Page 5)
  3. Title of the paragraph have been changed to ‘Skin pre-treatment chemicals.’ (line 175. Page 5)
  4. Interaction of iodine treatment and the activity of the skin has corrected to better reflect on the explanation which can be found in line 188-195. Page 6
  5. The role of SC and the electrical resistance of the skin has been added for better understanding which can be found in line 206-220. Page 6
  6. Specific lipids used have been added in line 236-239. Page 6
  7. Reference to line 279 (previously 219) have been added accordingly. References no 58 (previously 56). Page 7
  8. Figure on isopropyl myristate; polyvinyl alcohol and glutaraldehyde have been added as figure 4. (Line 314. Page 8)
  9. Molecular structures of lecithin and ginsenoside have been added as figure 5 (line 363. Page 9)
  10. The word cholesterol has been replaced with steroids (line 355. Page 9)
  11. Explanation on the differences between skin of rodents and human have been added (line 418-423. Page 11)

Note: the page or line may slightly differs due to the conversion of word to pdf

Round 2

Reviewer 2 Report

This revision does not substantially improve the manuscript to the point that it is worthy of publicatio as it will be of minimal value to the field of transdermal delivery or indeed diabetes management. The manuscript is poorly constructed with much text devoted to very well known information about diabetes management, insulin, skin permeation per se and general methods to enhance permeation. It is not particularly up to date, lacks critical insight, and therefore does not provide a useful critical, up to date, insightful and well written review.

Reviewer 3 Report

Dear authors,

I checked the revised manuscript and I realized that you corrected according with my suggestions.

Kind regards